# ReLU Strikes Back: Exploiting Activation Sparsity in Large Language Models

**Iman Mirzadeh[†], Keivan Alizadeh, Sachin Mehta, Carlo C Del Mundo[◇]**
**Oncel Tuzel, Golnoosh Samei[◇], Mohammad Rastegari, Mehrdad Farajtabar[†]**
Apple

## Abstract

Large Language Models (LLMs) with billions of parameters have drastically transformed AI applications. However, their demanding computation during inference has raised significant challenges for deployment on resource-constrained devices. Despite recent trends favoring alternative activation functions such as GELU or SiLU, known for increased computation, this study strongly advocates for reinstating ReLU activation in LLMs. We demonstrate that using the ReLU activation function has a negligible impact on convergence and performance while significantly reducing computation and weight transfer. This reduction is particularly valuable during the memory-bound inference step, where efficiency is paramount. Exploring sparsity patterns in ReLU-based LLMs, we unveil the re-utilization of activated neurons for generating new tokens and leveraging these insights, we propose practical strategies to substantially reduce LLM inference computation up to three times, using ReLU activations with minimal performance trade-offs.

## 1 Introduction

The widespread excitement surrounding Large Language Models (LLMs) has sparked significant interest in leveraging AI across diverse domains (Brown et al., 2020; Chowdhery et al., 2022; Bubeck et al., 2023). However, realizing the potential of LLMs is challenged by their significant computational and memory requirements during inference (Pope et al., 2023; Kim et al., 2023c; Aminabadi et al., 2022). To enhance the inference efficiency[1], various techniques have been explored, including quantization (Dettmers et al., 2022; Liu et al., 2023a), speculative decoding (Kim et al., 2023d), pruning (Ma et al., 2023; Sun et al., 2023), and weight sparsification (Frantar & Alistarh, 2023; Dong & Chen, 2023). Among these techniques, achieving activation sparsity offers a compelling advantage by providing a favorable balance between accuracy and speedup, especially on modern hardware like GPUs (Liu et al., 2023b).

Notably, employing the Rectified Linear Unit (ReLU) activation function (Fukushima, 1969) in neural networks is recognized for inducing sparse activations and has been adopted in various prior works (He et al., 2016; Kurtz et al., 2020; Li et al., 2023; Sheng et al., 2023). To reaffirm this property, we employ the OPT model (Zhang et al., 2022a), utilizing ReLU, and measure the sparsity of activations in the Feed Forward Network (FFN) between the fully connected layers. As illustrated in Fig. 1a, all layers exhibit sparsity exceeding $90\%$. On average, across all layers, this activation sparsity results in substantial weight transfer (I/O) savings between the GPU and CPU, impacting $95\%$ of the rows of the down projection layer's weights (Fig. 1b). This reduction directly translates to computation savings, as for these rows, the result of the matrix multiplication operation will be zero. Furthermore, unlike unstructured sparsity (e.g., weight pruning), this type of sparsity is more hardware-friendly due to zeroing more extensive and structured chunks, such as rows or

---

[†]Corresponding authors: {imirzadeh, farajtabar}@apple.com

[◇]Work done at Apple.

[1]In this work, we use FLOPS as a proxy for inference efficiency. In Appendix B, we demonstrate that for LLMs with activation sparsity, FLOPS can serve as a good approximation of real-world efficiency due to the structure inherent in activation sparsity (e.g., skipping the entire row corresponding to zero activations).

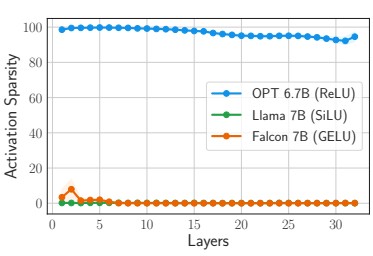 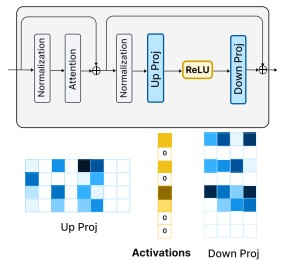 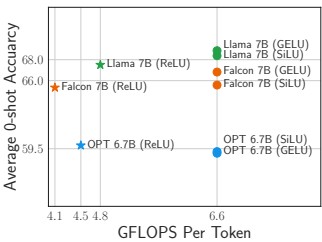

(a) Sparsity of different models    (b) Sparsity for Efficiency    (c) Accuracy vs. Computation

Figure 1: **(a)** Activation Sparsity of different pretrained models: ReLU-based OPTs show significantly higher sparsity. **(b)** Zeroed out entries after ReLU save compute in large semi-structured chunks (e.g., rows). **(c)** Comparison of inference efficiency and performance of the different models with different activation functions after fine-tuning: The choice of activation function does not significantly impact the accuracy, as any of GELU, SiLU, or ReLU can be used on all three models and achieve the same level of accuracy as the original activation function. However, using ReLU can provide an additional benefit of leading to activation sparsity and faster inference.

columns (Jaszczur et al., 2021; Liu et al., 2023b). For OPT models, this sparsity reduces the computation required for inference from 6.6G FLOPS (Floating Point Operations Per Second) to 4.5G FLOPS per token, resulting in a 32% computation saving (Fig. 1c).

However, a recent trend has emerged, favoring variations of ReLU that are smoother but more complex (Hendrycks & Gimpel, 2016; Ramachandran et al., 2017). These alternatives have gained popularity due to their slightly faster convergence and improved final accuracy (Shazeer, 2020). For example, PaLM (Chowdhery et al., 2022) and Llama models (Touvron et al., 2023) adopt SiLU[2] (Hendrycks & Gimpel, 2016; Elfwing et al., 2018; Ramachandran et al., 2017), while MPT (MosaicML, 2023) and Falcon models (Almazrouei et al., 2023) use GELU (Hendrycks & Gimpel, 2016). Nonetheless, as demonstrated in Fig. 1c, when we finetune several pretrained LLMs with different activation functions, their performance does not change significantly (within a specific model), while ReLU models require much less computation.

In this paper, we re-evaluate using ReLU for LLMs. We are motivated by the pragmatic consideration that, in many real-world applications and computational platforms capable of supporting sparse vector-matrix multiplications, computational efficiency during *inference* outweighs the one-time computational cost incurred during training. We make the following contributions:

- We demonstrate that when trained from scratch, there is no significant difference in terms of performance between different activation functions. However, in terms of computational requirements during inference, ReLU activations prove significantly lighter (Sec. 3).

- Considering that many modern LLMs (e.g., Llama and Falcon) have been trained with non-ReLU activations, and it is not cost-effective to train them from scratch, we investigate fine-tuning these models with ReLU activations. We show that the models quickly regain their original performance across various reasoning and reading comprehension tasks (Sec. 4.1). Moreover, we show that by leveraging the activation sparsity of ReLU layers and inserting additional ReLU layers after normalization layers, we can further reduce inference FLOPS by up to threefold (Sec. 4.2).

- In addition to their computational benefits, we present two promising applications of activation sparsity that can inspire future work. Firstly, we demonstrate that LLMs with ReLU activations reuse a significant portion of already activated neurons during token generation, a phenomenon we term *aggregated sparsity* (Sec. 5.1). This reusability leads to an inference speedup for speculative decoding (Sec. 5.2). Additionally, we show that studying the pre-activations of pretrained LLMs can guide the selection of unconventional activation functions (e.g., *shifted ReLU*), leading to the increased sparsity while maintaining performance similar to ReLU activation (Sec. 5.3).

  Overall, we believe our work represents a significant step toward leveraging the potential of sparse activation functions for faster and more efficient inference in large language models.

---

[2]To be more precise, the mentioned models use SwiGLU activation function, but in this work, we focus on the gating projection module in the FFN that uses SiLU (Swish) activation function.

## 2 RELATED WORKS

**Activation Functions in Transformers.** The original Transformer architecture (Vaswani et al., 2017) was proposed with the ReLU activation function (Fukushima, 1969), following the popularity of ReLU at the time. Later, several studies aimed to improve the ReLU activation function by increasing its smoothness (Hendrycks & Gimpel, 2016) and/or including parameterized gating mechanisms, such as GELU, SiLU, GLU, and SwiGLU (Dauphin et al., 2017; Ramachandran et al., 2017). Earlier studies demonstrated the benefits of these alternatives to ReLU for transformers (Shazeer, 2020; Narang et al., 2021), but on a small scale (e.g., they trained models up to a couple of 100M parameters with at most 35B tokens, while in this work, we train 1B parameter models on more than 100B tokens). However, we believe the impact of activation functions on performance is marginal, following scaling laws (Kaplan et al., 2020; Hoffmann et al., 2022), which state that architectural changes do not significantly impact performance.

**Activation Sparsity.** Existing research shows increased sparsity reduces inference and training times (Kurtz et al., 2020; Han et al., 2023; Song et al., 2021; Zhang et al., 2022b; Li et al., 2022; Liu et al., 2023b). For instance, Jaszczur et al. (2021) uses ReLU and added a controller to both promote and predict sparsity, while other works only use prediction modules to predict the activation masks (Liu et al., 2023b). We note that the mentioned works assume the pretrained model has already been using a sparse ReLU activation, and hence, only training a separate module to predict sparsity could be enough. However, we note that most LLMs pretrained these days do not use ReLU, and we aim to bridge this gap. Moreover, these works focus only on a single transformer architecture while we focus on various architectures so our findings can be practical. Finally, we show that there is no need to train a separate prediction module that complicates the computation graph, and using efficient ReLU layers can be enough.

**Speculative Decoding and Sparsity.** Speculative decoding combats latency under memory constraints using a smaller model for token prediction and a larger model for verification (Leviathan et al., 2023; Kim et al., 2023d). Investigating its integration with sparsity, we find activation sparsity exhibits a temporal pattern, enhancing speculative decoding. We provide guidelines for parameter selection when incorporating sparsity.

We defer other lines of related works that are orthogonal to our work, such as model compression techniques, sparse attention methods, and Mixture of Experts (MoE) to Appendix A.

## 3 DOES THE ACTIVATION FUNCTION IMPACT PERFORMANCE?

This section first overviews our experimental setup, including models, data, and evaluations. Then, by training various models from scratch with different activation functions, we demonstrate that changing activation functions minimally impacts performance. However, the impact on inference efficiency is substantial.

### 3.1 EXPERIMENTAL SETUP

**Models.** We use open source pretrained models such as OPT (Zhang et al., 2022a), Llama (v1) (Touvron et al., 2023), and Falcon (Almazrouei et al., 2023) as they use different architectures and pretraining setup (e.g., attention/FFN structure/normalization, activation functions), allowing our study covers a wider range of models.

**Datasets.** We use the RefinedWeb dataset (Penedo et al., 2023), for our pretraining in Sec. 3.2 and finetuning pretrained models in Sec. 4. We chose RefinedWeb because it is a high-quality subset of Common Crawl, which is often used in the pretraining phase of LLMs, including Llama, Falcon, and OPT. We also use the validation split of WikiText (Merity et al., 2017) for measuring the sparsity and recording preactivation distributions of various pretrained models. However, our conclusions hold for other datasets we have tested.

**Training and Finetuning.** For finetuning the pretrained models, we follow the original pretraining recipe, except we use a fixed learning rate of 1.5e-5 for Llama 7B, Falcon 7B, and OPT 6.7B models. In addition, we use the AdamW optimizer (Loshchilov & Hutter, 2019) for our finetuning with ZeRO stage 1 (Rajbhandari et al., 2020), where we shard the optimizer states across different GPUs. For pretraining OPT 1.3B models from scratch in Sec. 3.2, we follow the OPT training recipe.

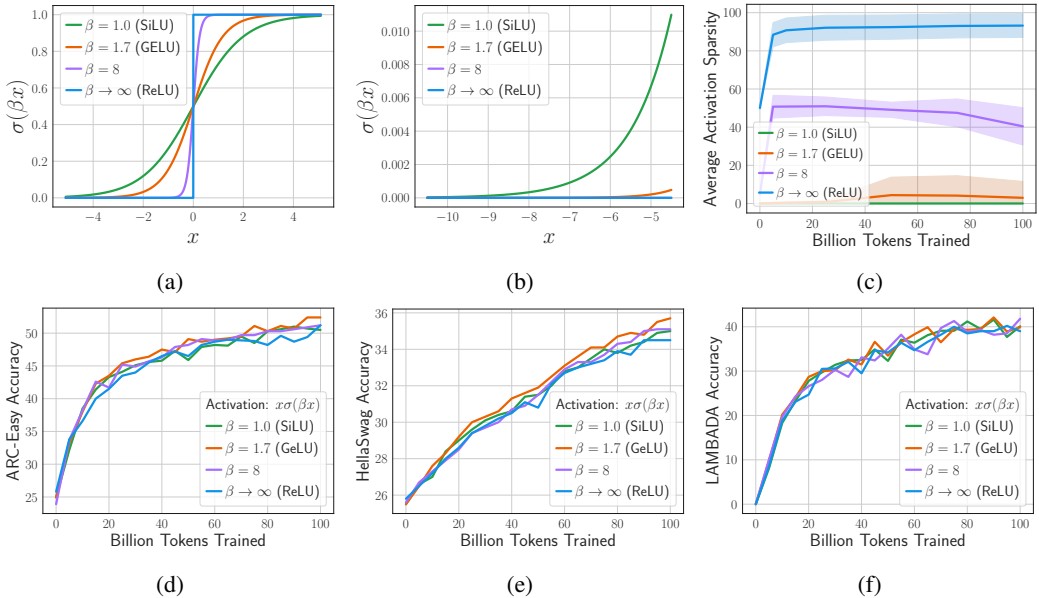

(a)          (b)          (c)

(d)          (e)          (f)

Figure 2: **(top)** (a) Shapes of different gating functions over [-5, 5]; (b) Continuation of (a) where SiLU is comparably larger compared to others; (c) Sparsity of the FFN with different activations: increasing $\beta$ increases sparsity. **(bottom)** when trained from scratch, OPT 1.3 B models using different activation functions achieve similar performance.

**Evaluation.** For our *performance* evaluation, we use the few-shot tasks from Language Model Evaluation Harness (Gao et al., 2021). We select these tasks such that they can measure various abilities of the models (e.g., reading comprehension, reasoning, etc.), and we aim to be consistent with other works in the literature to make the comparison easier. Consistent with the other sections, we compare activation sparsity as a measure of *efficiency*. Further details regarding the relationship between activation sparsity, FLOPS, and inference efficiency are discussed in Appendix B.

## 3.2 TRAINING FROM SCRATCH: PERFORMANCE AND SPARSITY

While the previous literature suggests that non-ReLU variants can improve the performance of transformers (Shazeer, 2020; Narang et al., 2021), we argue the impact is marginal at best. To support our claim, we train the OPT 1.3B model from scratch on a hundred billion tokens of the RefinedWeb datasets with different activation functions, including ReLU, SiLU, and GELU. All these activation functions can be viewed as f(x) = $x \cdot \sigma(\beta x)$, where $\beta$ controls the gating part (smoothed cutoff threshold) of the activation function (see Fig. 2a). For $\beta = 1$, we will have SiLU($x \cdot \sigma(x)$), and $\beta = 1.7$ is a good approximation of GELU. Finally, as $\beta \to \infty$, the activation function becomes closer to ReLU. To further explore the spectrum of ReLU to SiLU we add another one with $\beta = 8$.

As shown in the bottom row of Fig. 2, the performance of the models is very similar when using different activation functions. This is consistent with the scaling laws literature ((Kaplan et al., 2020; Hoffmann et al., 2022)), which suggests that the performance of sufficiently large models trained on sufficiently large data depends heavily on compute and data, not architectural details.

While the performance levels of the different activations are similar, their activation sparsity levels differ. Here, we define sparsity as the average sparsity level across all layers for each model. As shown in Fig. 2c, as we transition from SiLU to ReLU (increasing $\beta$), the sparsity also increases. This results from the different gating thresholds, as ReLU drops significantly more values compared to GELU and SiLU (see Fig. 2b). In Appendix D, we illustrate the evolution of the pre-activation distribution throughout training.

Overall, the results support our initial claim: non-ReLU activations result in a negligible performance gain (if any) but a substantial loss in sparsity and efficiency. While, at times, the performance of GeLU or SiLU might be slightly higher, ReLU can match it with slightly longer training. We acknowledge that to compensate for the small gap in performance, we need to pay the one-time cost of longer training. However, in return, we get a significantly more sparsity.

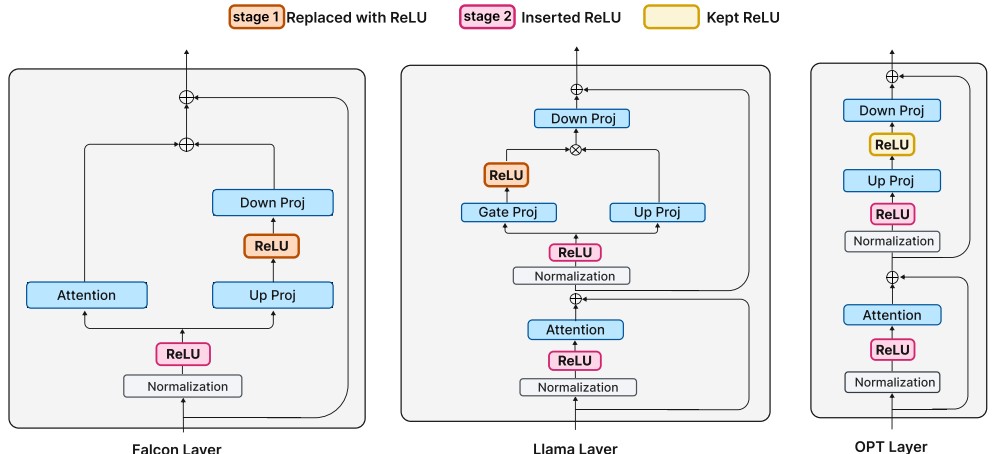

Figure 3: Architectural surgeries for *relufication*. In stage 1 we keep the existing ReLUs (in the case of OPT) or replace the activation function between up projection and down projections from GELU (Falcon) and SiLU (Llama) to ReLU. In stage 2, we insert new ReLUs after normalization layers.

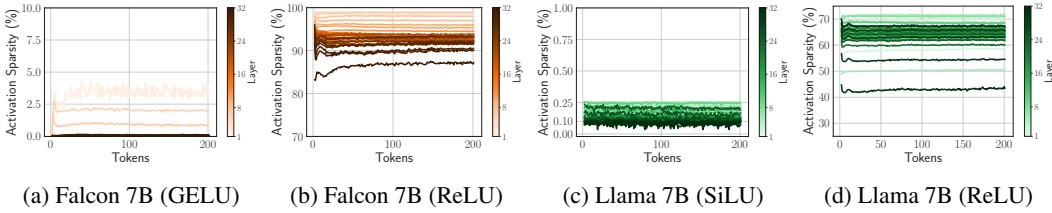

|                       |                       |                      |                      |
|:---------------------:|:---------------------:|:--------------------:|:--------------------:|
| (a) Falcon 7B (GELU)  | (b) Falcon 7B (ReLU)  | (c) Llama 7B (SiLU)  | (d) Llama 7B (ReLU)  |

Figure 4: Activation sparsity of Falcon and Llama models improves significantly after *relufication*.

## 4 RELUFICATION

While in the previous section, we have seen that the performance does not depend on the activation function, we note that most of the available pretrained LLMs are trained with activation functions other than ReLU. Hence, to incorporate the computational benefits of ReLU activations at inference time, we perform various architectural surgeries and study the consequences of such changes.

We present our findings about incorporating ReLU activations into the pretrained LLMs, a process we refer to as *relufication*. More specifically, we show that replacing the activation functions of pretrained LLMs with ReLU is possible, and the performance can be recovered very rapidly during finetuning. Moreover, we show that we can exploit the sparse ReLU activations, and by inserting additional ReLU layers after normalization layers, we can improve inference efficiency, as FLOPS indicates. Finally, we show these modifications, which are easy to implement, lead to lighter models at inference time while maintaining comparable performance to the original pretrained models.

### 4.1 STAGE 1: REPLACING NON-RELU ACTIVATIONS

The process of relufication for different pretrained architectures is shown in Fig. 3. This process can be done in multiple stages, as we describe here. The first and more intuitive stage replaces non-ReLU activations with ReLU in the FFN layer. For the Falcon and Llama models, this means replacing GELU and SiLU, respectively. We note that since OPT models already use ReLU activations, we keep those unchanged. After finetuning on 30 billion tokens of the RefinedWeb, Fig. 4 shows that the modified models have significantly more sparsity in their activations.

In addition to the drastic improvement in activation sparsity, we can make several notable observations. First, while the shape of preactivation depends on the pretraining dynamics and architecture, in Fig. 5, we show that it does not change significantly during the relatively short finetuning stage.

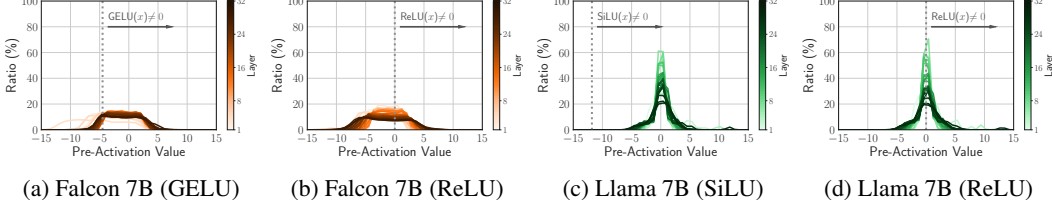

| (a) Falcon 7B (GELU) | (b) Falcon 7B (ReLU) | (c) Llama 7B (SiLU) | (d) Llama 7B (ReLU) |

Figure 5: The preactivation distribution of pretrained models for Falcon and Llama does not change significantly during the short finetuning stage of relufication. The dashed line shows the cutoff point before which the output is almost zero.

As a result, we can predict the activation sparsity before finetuning, knowing it will not change significantly. Later in Sec. 5.3 we build on this observation and propose shifting the preactivation values before applying ReLU and further increasing the activation sparsity. The stability of the pre-activation distribution may suggest that the behavior of the network does not change while creating sparse representations. Indeed, we show that after replacing the activation function with ReLU, finetuned models quickly recover their performance in Fig. 6. We believe optimizing this process even further (e.g., using better finetuning data) is an exciting follow-up direction.

## 4.2 STAGE 2: PUSHING FOR MORE SPARSITY

In the previous stage, we replaced non-ReLU activations to gain more sparsity. This leads to the input of *down projection* layer being sparse, roughly 30% of the total computation. However, there are other matrix-vector multiplications in the decoder layer of transformers besides the down projection. For instance, before the *up projection* and *gate projections* of FFN layer, and *QKV projections* in the attention layer (see Fig. 3). Together, the mentioned matrix-vector multiplications consume about 55% of the total computation.

To this end, we utilize the fact that in modern transformer layers, the input to both the attention and FFN layers come from a normalization layer, e.g., LayerNorm (Ba et al., 2016) or RMSNorm (Zhang & Sennrich, 2019). These layers can be viewed as a specific form of MLP, where, instead of applying arbitrary learnable parameters, they learn to scale inputs. Therefore, we apply ReLU to obtain sparse activations after normalization layers which we call the *second stage* of relufication in Fig. 3.

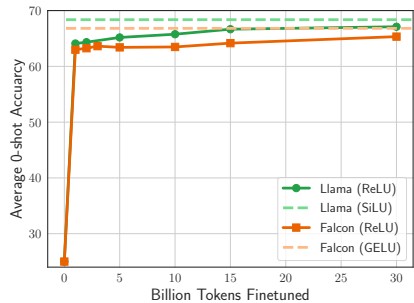

Figure 6: Evolution of zero-shot accuracy during finetuning: The model quickly recovers most of its lost performance due to the architecture surgery.

Tab. 1 shows that different stages of the relufication process do not significantly reduce zero-shot accuracy while using significantly less compute. The sparsity is broken down into three categories: up, down, and QKV projections. Notably, the input to QKV is less sparse than FFN projections, which opens an interesting avenue for future research. We note that the small gap in performance between the original vs. relufied models may be partially due to the finetuning process and not necessarily the activation function. Our finetuning is applied only for 30B and 50B tokens for stages 1 and 2, respectively. Putting into prospect and comparing it with 1T tokens of Llama, for example, this is equivalent to 3-5% of the original training duration. As discussed in Sec. 3.2, according to the scaling properties of LLMs, the gap will be further bridged by additional finetuning steps.

We also assess the in-context learning ability of the relufied models with the Massive Multitask Language Understanding (MMLU) (Hendrycks et al., 2021) benchmark in Tab. 2. Our results show that when we augment the original LLMs with different activations and finetune, the few-shot performance does not change significantly either. Moreover, Sec. E in the appendix shows that a larger but relufied model performs better than an original smaller model of the same FLOPS. Overall, the results affirm that the proposed relufication procedure can decrease the inference FLOPS at various stages and rates while maintaining on-par performance on various tasks.

Table 1: Comparing zero-shot performance across several tasks: After *relufication*, the activation sparsity of models increases significantly, hence increased efficiency measured by FLOPS. Within each group, the performance levels are comparable.

| Model (stage) | Input Sparsity (%) | | | FLOPS (G) | Zero-Shot Accuracy (%) | | | | | | | | | |
|---|---|---|---|---|---|---|---|---|---|---|---|---|---|---|
| | QKV | DownProj | UpProj | | Avg | Arc-E | Arc-C | Hellaswag | BoolQ | PIQA | LAMBADA | TriviaQA | WinoGrande | SciQ |
| OPT 1.3B | 0 | 96 | 0 | 1.3 | 50.7 | 57.3 | 22.9 | 41.3 | 57.0 | 71.8 | 56.0 | 6.1 | 58.9 | 84.6 |
| OPT 2.7B (s2) | 50 | 96 | 35 | 1.1 | 53.1 | 60.3 | 26.8 | 44.9 | 55.4 | 73.9 | 57.6 | 12.4 | 59.6 | 86.7 |
| OPT 2.7B | 0 | 96 | 0 | 1.8 | 54.5 | 63.3 | 29.2 | 45.8 | 57.6 | 74.2 | 61.4 | 12.3 | 60.8 | 85.9 |
| OPT 6.7B (s2) | 50 | 97 | 40 | 2.8 | 58.6 | 66.5 | 32.2 | 49.1 | 63.0 | 76.4 | 63.3 | 23.8 | 63.1 | 90.3 |
| OPT 6.7B | 0 | 97 | 0 | 4.5 | 59.8 | 68.0 | 32.4 | 50.2 | 68.4 | 75.8 | 67.2 | 20.9 | 65.3 | 90.2 |
| Falcon 7B (s2) | 56 | 95 | 56 | 2.2 | 64.8 | 73.6 | 38.6 | 55.3 | 68.4 | 78.9 | 67.6 | 40.4 | 67.1 | 93.4 |
| Falcon 7B (s1) | 0 | 94 | 0 | 4.1 | 65.2 | 72.2 | 39.1 | 55.4 | 70.6 | 78.4 | 69.2 | 40.5 | 67.5 | 93.1 |
| Falcon 7B | 0 | 1 | 0 | 6.6 | 66.8 | 74.6 | 40.2 | 57.7 | 73.5 | 79.4 | 74.5 | 40.4 | 67.2 | 94.0 |
| Llama 7B (s2) | 51 | 65 | 67 | 2.9 | 66.4 | 73.8 | 39.6 | 54.8 | 69.9 | 77.9 | 70.7 | 48.5 | 68.6 | 93.8 |
| Llama 7B (s1) | 0 | 62 | 0 | 4.8 | 67.1 | 75.2 | 40.1 | 55.2 | 73.4 | 77.7 | 71.5 | 49.6 | 67.1 | 94.2 |
| Llama 7B | 0 | 0 | 0 | 6.6 | 68.4 | 75.5 | 42.1 | 56.9 | 74.8 | 78.7 | 73.1 | 49.9 | 69.8 | 95.4 |

Table 2: MMLU five-shot accuracy. Models finetuned with different activation functions have similar performance.* Denotes we replace the SiLU function in Llama's SwiGLU activation function with ReLU.

| Model | Activation | FLOPS(%) | Avg | Humanities | STEM | Social Sciences | Other |
|---|---|---|---|---|---|---|---|
| Falcon 7B | SiLU | 100 | 26.4 | 24.8 | 27.4 | 27.2 | 26.2 |
| Falcon 7B | GELU | 100 | 27.7 | 28.1 | 26.0 | 28.0 | 29.4 |
| Falcon 7B | ReLU | 62 | 27.9 | 26.0 | 26.5 | 31.8 | 27.9 |
| Llama 7B | SiLU* | 100 | 35.1 | 37.9 | 30.2 | 37 | 37.1 |
| Llama 7B | GELU | 100 | 35.9 | 38.4 | 29.4 | 37.6 | 39.5 |
| Llama 7B | ReLU | 72 | 34.7 | 34.8 | 31.2 | 36.3 | 37.8 |

## 5 APPLICATIONS

In this section, we discuss promising directions motivated by our investigation in Sec. 4. First, we introduce *aggregated sparsity*, showing that ReLU networks reuse previously activated neurons when generating tokens. Hence, we can leverage this to increase the generation speed. Next, we relate aggregated sparsity with speculative decoding to further improve speculative decoding's inference time. Finally, we briefly discuss a promising direction of using the *shifted ReLU* activation function to improve the sparsity further.

### 5.1 AGGREGATED SPARSITY: REUSING PREVIOUSLY ACTIVATED NEURONS

A consequence of using only a small subset of neurons for each token is that if these neurons are shared to some degree, the model still does not use all of the neurons until many tokens are processed. We refer to this as *aggregated sparsity*, which we defined as the ratio of neurons that have not been used up to processing the first $t$ token. Note that this metric will always be non-increasing. Intuitively, it measures the unused capacity of feed-forward neurons for processing a specific prompt.

Here in Fig. 7a we show that for the OPT-6.7B model, on average, about 50% of all the neurons will be unused across the first 150 tokens of prompts coming from the WikiText dataset. Our empirical results hold for other ReLU models and other datasets. Additionally, in Fig. 7b, we show that this pattern is far from random activation of neurons during the token generation with a rate equal to the average rate of activation usage per token. Let $s_i$ be the activation sparsity of layer $i$ averaged over all tokens. Then, the probability of an activation not used in generating the first $t$ tokens in uniformly random selection is $s_i^t$. Fig. 7b shows this quantity for two layers $i = 8, 24$ for the first $256$ tokens in dashed line. It also shows the real (observed) number of activations being used in the solid line. The fact that the random aggregated sparsity (referred to as random sparsity) is lower than the observed aggregated sparsity (we refer to it as aggregated sparsity) shows a clear pattern of reusing activations.

We can benefit from the overlapping activations by utilizing previously loaded weights from the down projection layer for upcoming tokens.[3] To test this, we initiate with reading 128 tokens. For

---

[3]Although we can assume in some cases the weights are already in the GPU's RAM, there is an additional copy cost from GPU memory to cache and registers that can be reduced if the neuron reuse (sharing) happens across different generation steps. In addition, for the cases where we use offloading/checkpointing, there's an additional I/O overhead for copying from CPU to GPU memory and our analysis can be applied.

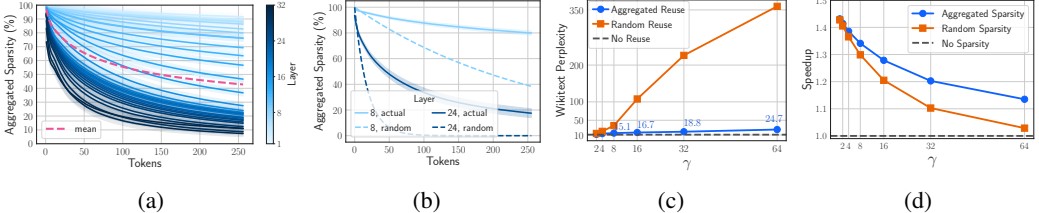

Figure 7: **(a)** Aggregated sparsity of different layers and their mean. **(b)** Aggregated sparsity during token generation and comparison with a random sparsity. **(c)** Perplexity, based on the number of tokens for which loaded weights from previous tokens are reused. The dashed line represents no reuse, the solid blue line shows the case with activation reuse according to aggregated sparsity, and the orange line depicts the perplexity when activations are reused according to a random sparsity. **(d)** The inference speedup of speculative decoding with aggregated sparsity and with random sparsity. Speedup equal to 1.0 is the standard version of speculative decoding.

the subsequent 128 tokens, we intermittently avoid loading new weights for every $\gamma$ token. Using $\gamma = 16$ as an example, tokens 129-145 are generated conventionally. However, for tokens 146-161, we retain the existing weight without introducing any new weight. This pattern continues, with every next set of $\gamma$ tokens alternating between conventional generation and weight reuse. In Fig. 7c, we observe only a slight increase in perplexity when using this approximation to address the memory and I/O-intensive nature of LLM inference. This figure contrasts the perplexity obtained from reused activations and random selections. The reuse strategy aligns well with the baseline, whereas random selection notably increases perplexity, highlighting the effectiveness of reusing the already loaded activations for subsequent tokens.

## 5.2    ACTIVATION SPARSITY AND SPECULATIVE DECODING

As highlighted in Sec. 5.1, activation reuse happens for multiple consecutive tokens. When multiple consecutive tokens are processed together, we can save the I/O (i.e., transferring weights to GPU/CPU as discussed in Appendix B) associated with activations that are not used in any of them. If the reuse was not happening, and the sparsity of all tokens was purely random, the aggregated sparsity would shrink exponentially and quickly diminish. Speculative decoding (Leviathan et al., 2023) is a related technique that uses a smaller model $M_q$ to propose $\gamma$ tokens and a larger model $M_p$ to verify those tokens and select matching ones. It improves the runtime of the model by avoiding running $M_p$ sequentially.

To improve speculative decoding, aggregated sparsity can trim down the portion of the model that needs to be run. Instead of running the full model, only the non-sparse parts need to be evaluated, which will reduce I/O and compute latency. Suppose the average aggregated sparsity of $M_p$ for $\gamma$ tokens is $\bar{s}_{\text{agg}}(\gamma)$, and cost of running $M_q$ over $M_p$ is $c$. Then the expected latency speedup when going from standard speculative decoding to sparse speculative decoding is $\frac{c\gamma+1}{c\gamma+(1-\bar{s}_{\text{agg}}(\gamma))}$.

Fig. 7d compares sparse speculative decoding to the standard version for OPT 6.7B model. As a case study, for $\gamma = 16$, the sparse version has a 1.27x speedup over the standard speculative decoding. If the aggregated sparsity was random over different tokens, the speedup would have been only 1.20x. Note that even random sparsity will lead to speedup over standard speculative decoding. This further shows the value of relufication. However, the speedup due to random sparsity would diminish quickly in comparison to aggregated sparsity as we go for larger $\gamma$. For example, for $\gamma = 64$ the speedup is almost negligible, while the speedup for the aggregated sparsity is around 1.14x. Further discussion and details are postponed to Appendix C, where we compare sparse speculative decoding, standard speculative decoding, and autoregressive decoding and discuss optimal $\gamma$ in the case of sparse speculative decoding.

## 5.3    THE SHIFTED RELU ACTIVATION

Our work in this section is motivated by the observation from Sec. 4, where, comparing Fig. 4d with Fig. 4b revealed that the relufied Llama has much less sparsity (65%) than the relufied Falcon model

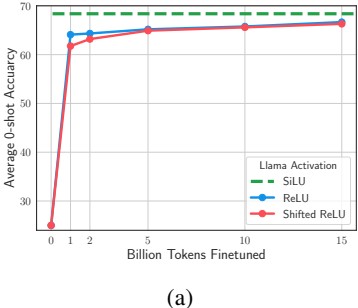 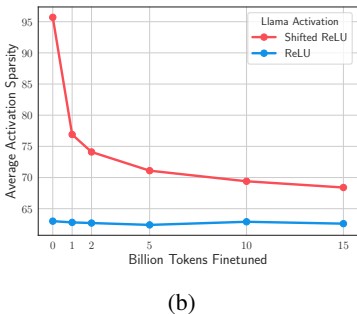

(a)                             (b)

Figure 8: The effect of shifted ReLU on Llama model. **(a)** The performance is almost the same as the original ReLU. **(b)** Shifted ReLU (i.e., $\text{ReLU}(x-1)$) is much sparser than the original ReLU.

(95%). In addition, we build on two of our previous findings. First, the preactivation distribution of the relufied Llama (Fig. 5c) includes a considerable mass after the cutoff value at zero. Second, the shape of the preactivation distribution does not change before and after the relufication process (Fig. 5c and Fig. 5d). Therefore, we may be able to shift the preactivation distribution to the left to put more volume before the cutoff at 0. Another benefit of this approach is simplicity, as this does not require changing the loss function or the training regime.

To this end, for preactivation input $x$, rather than applying $\text{ReLU}(x)$, we use $\text{ReLU}(x - b)$ where $b \in \mathbb{R}$ is a constant scalar. We propose to set the value $b$ based on the preactivation distribution. For instance, based on the distribution in Fig. 5d, setting $b = 1$ and hence using $\text{ReLU}(x - 1)$ as our activation function will result in initially dropping 95% of the preactivations and make it significantly sparser.

Figure 8a shows that the shifted ReLU activation function has on-par accuracy with the ReLU activation function. Moreover, similar to our observation in Sec. 4, the shifted ReLU activation quickly recovers the lost performance due to the drastic change of activation function, while it also maintains a very high-level activation sparsity during the finetuning stage. Finally, an important observation in Fig. 8b that as the finetuning process progresses, the sparsity level of the model with shifted ReLU quickly decreases[4].

A deeper investigation of ReLU-variants that can promote sparsity without sacrificing performance is an appealing future direction. Moreover, it will be interesting to study the impact of the shifted ReLU for stage-2 of our relufication process where the sparsity level is usually not very high.

## 6 CONCLUSION

In this study, we conducted a large-scale investigation of the activation functions, and we have shown that the choice of activation functions during pretraining and finetuning does not have a significant impact on performance while using ReLU can provide an additional benefit of leading to activation sparsity and more efficient inference. To bridge the gap between existing pre-trained models and our work, we have *relufied* several models to incorporate the ReLU activation function into the architecture of these already pre-trained models. We have shown that across several zero-shot and few-shot tasks, the ReLU-based LLMs perform similarly to their non-ReLU models at a significantly reduced computation. In addition, after observing sparsity patterns in ReLU LLMs, we explored a few promising directions to improve the token generation speed through *aggregated sparsity* and achieve greater efficiency using ReLU-based activation functions like *shifted ReLU*.

We believe our work is among the few studies that investigate changes in the architectural components of LLMs on a large scale. We hope our findings motivate the community to further investigate the advantages of well-structured activation sparsity, ultimately enhancing the efficiency of these models.

---

[4]During the camera-ready preparation, we discovered an implementation error in our code that resulted in incorrect sparsity calculations for our shifted-ReLU model. Consequently, we have updated Figure 8b.

ACKNOWLEDGEMENT

The authors would like to thank Fartash Faghri, Minsik Cho, Thomas Merth, Mohammad Samragh, Moin Nabi, Arsalan Farooq, and Peter Zatloukal for their invaluable discussions and support.

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

APPENDIX

## A  EXTENDED RELATED WORKS

**Activation Functions.** ReLU, introduced by (Fukushima, 1969), remains a predominant activation function for deep neural networks and was notably utilized in the original transformers work (Vaswani et al., 2017). SwiGLU (Shazeer, 2020) has been shown to enhance performance when replacing ReLU in feedforward layers and is a feature in models like Llama (Touvron et al., 2023). Narang et al. (2021) conducted an extensive comparison of various activation functions, such as GeLU, SiLU (Hendrycks & Gimpel, 2016; Elfwing et al., 2018), ELU (Clevert et al., 2016), SeLU (Klambauer et al., 2017), and GLU variants (Dauphin et al., 2017), identifying certain advantages over ReLU. Our paper and results differ from theirs by training billion scale models and data as opposed to their smaller scale ones. Furthermore, our results indicate that extended training can diminish the performance gap between ReLU and these other functions, also leading to savings in computational costs.

**Activation Sparsity.** A body of prior research (Kurtz et al., 2020; Han et al., 2023; Song et al., 2021) has demonstrated that increasing sparsity can lead to reductions in both inference and training times. Dejavu (Liu et al., 2023b) and Li et al. (2022) observed pronounced sparsity in activations when using the ReLU function in feedforward layers. These studies propose that predicting this sparsity can further boost inference speeds. Similarly, Jaszczur et al. (2021) employed ReLU activations and introduced a controller to actively promote sparsity. Notably, these studies predominantly focus on networks employing ReLU activations, leaving out those with alternative activation functions. In contrast, our approach modifies networks by substituting other activation functions with ReLU. We then fine-tune these networks to achieve activation sparsity in the MLP layer post-ReLUfication. We further illustrate that inserting ReLU prior to the QKV and Feedforward layers can substantially reduce FLOPS, albeit at a minor cost to accuracy. Unlike the aforementioned studies, we do not utilize a sparsity predictor to minimize FLOPS.

**ReLU in Attention Mechanisms.** Beyond the activation function in the MLPs of large language models, a softmax activation is often employed within the attention module. Prior studies have indicated that it's feasible to replace this softmax with ReLU without compromising accuracy (Wortsman et al., 2023; Shen et al., 2023; Hron et al., 2020). This avenue of research is distinct from our approach of Relufication, which specifically focuses on activations preceding weight multiplications.

**Model compression for efficient inference** Quantization, pruning and distillation are the main three techniques for compressing neural networks (Zhu et al., 2023). Quantization has been used to reduce model size and faster inference (Dettmers et al., 2023a; Liu et al., 2023a; Park et al., 2023; Dettmers et al., 2022; Lin et al., 2023; Lee et al., 2023; Dettmers et al., 2023b; Kim et al., 2023b; Chee et al., 2023; Xiao et al., 2023). The quantized model occupies less space reducing the memory latency (Frantar et al., 2022; Kim et al., 2023a). Reluification is orthogonal to quantization and reduces the amount of memory required to be loaded and can further decrease the memory latency. Distillation (Hsieh et al., 2023; Hinton et al., 2015; Gu et al., 2023; Mirzadeh et al., 2020; Agarwal et al., 2023) is another technique to train smaller models. This is orthogonal to using ReLU activations as any activation can be used in distillation methods. Sparsifying or pruning weights of neural networks (Frantar & Alistarh, 2023; Jaiswal et al., 2023; Zhang et al., 2023; Sun et al., 2023; Santacroce et al., 2023; Ma et al., 2023) can reduce computation and inference time. Weight sparsity is usually unstructured and hard to implement for hardware, but the sparsity induced by ReLU can easily be implemented as a matrix multiplication of non zero rows. Weight sparse models can be combined with our relufication for further decrease in compute.

**Mixture of Experts.** Mixture of Experts (MoE) LLMs usually subdivide the feed-forward layer into multiple experts. A router is then employed to selectively and sparsely activate these experts (Shazeer et al., 2017; Fedus et al., 2022b; Team et al., 2022). Similar to our work, MoE is a form of activation sparsity but in a group form and can be seen as a subset of sparse activation. Subsequent studies have further refined the inference and training methodologies for MoE models (Puigcerver et al., 2023; Hwang et al., 2023; Yi et al., 2023; Du et al., 2022; Kong et al., 2023; Rajbhandari et al., 2022; Zoph et al., 2022; Chen et al., 2022; Hazimeh et al., 2021). MoE can be also combined with Relufication, having sparsity inside FFN of each expert.

Another line of work is MoEfication of networks that have sparse activations by subdividing neurons (Zhang et al., 2022b). Relufication can also help MoEfication be applicable to a wider range of networks by increasing sparsity of FFNs. For a more in depth review of mixture of expert models we refer the reader to Fedus et al. (2022a).

**Speculative Decoding and Sparsity.** Speculative decoding is a method that aims to improve model latency when faced with memory bandwidth constraints Leviathan et al. (2023); Kim et al. (2023d). It involves using a smaller model to predict the next tokens, with a larger model subsequently verifying multiple tokens in a single operation. In this work, we examine the direct effects of incorporating sparsity into speculative decoding. We show that adding sparsity can lead to performance improvements in speculative decoding. Additionally, we provide guidelines on selecting parameters for speculative decoding when sparsity is introduced.

## B  DISCUSSION ON ACTIVATION SPARSITY AND INFERENCE EFFICIENCY

The primary motivation behind our work is to enhance *efficiency*, and we believe it is essential to provide a precise definition of this term. Throughout the main text, we predominantly use FLOPS as our efficiency metric. In this section, we argue why, in the presence of *activation sparsity*, FLOPS can serve as a suitable proxy for measuring various efficiency metrics.

Firstly, it is important to be reminded of the two major factors contributing to the efficiency of Large Language Models (LLMs): (1) the total amount of computation and (2) input/output (IO) transfer—i.e., transferring parameters from RAM to CPU/GPU for calculations. Notably, for today's large models, factor (2) acts as the major bottleneck during the inference phase. We refer the reader to the detailed analysis by Liu et al. (2023b). Ultimately, for a specific target device and assuming an efficient implementation, we believe that the most practical measure of efficiency is *latency* (e.g., the average time to generate a token). However, each device possesses its unique properties, necessitating a more ubiquitous proxy metric.

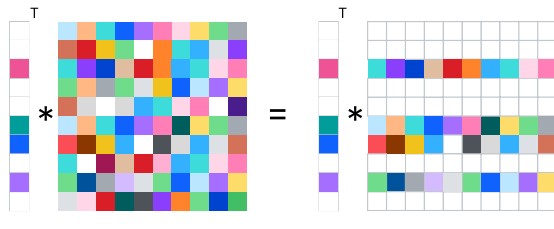

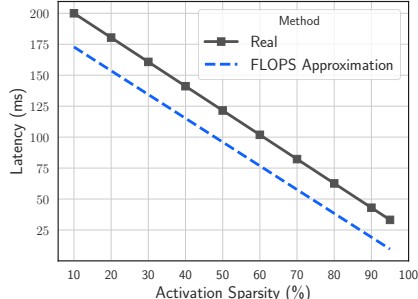

(a) sparse vector, dense matrix multiplication: by skipping rows, we reduce both the weight transfer (i.e., loading these rows for computation) and computation (i.e., the result will be zero).

(b) Comparing FLOPS versus real latency for OPT model (FFN).

Figure 9: For LLMs that have sparse activations, FLOPS is a good approximation of the real latency.

We argue that the way we calculated *FLOPS* in our paper and is greatly influenced by *activation sparsity* can reasonably approximate *efficiency*. Here are our reasons:

- **Reduced Computation**: As shown in Fig. 9a, with activation sparsity, we have a sparse vector-dense matrix multiplication at inference, while this will be a sparse-matrix-dense matrix multiplication during training. It is important to note that this is a semi-structured sparsity (unlike magnitude weight pruning), and we can leverage the fact that we are transferring weights in large chunks (i.e., *rows*). Modern accelerators already support sparse operations[5] and we can build on these existing tools.

- **Reduced IO Transfer**: During inference, weights need to be transferred to the device cache for computation (e.g., from RAM to CPU cache or GPU VRAM to GPU cache). This step constitutes

---

[5]For example, both cuSPARSE on NVIDIA CUDA® and Accelerate on Apple devices.

the main bottleneck during token generation. For instance, approximately 99.3% of the total latency is attributed to IO, as indicated by Liu et al. (2023b). However, by storing matrices in a row-major order, we can *skip loading* unnecessary rows as the output will be zero.

Overall, as depicted in Figure 9b based on the calculations by Liu et al. (2023b), we demonstrate that for the OPT model on an NVIDIA A100 node, counting FLOPS provides a reasonable approximation to and is highly correlated with the time needs to generate tokens, especially, for LLMs with activation sparsity.

### B.1 Dense-Matrix Sparse-Vector GeMV GPU Kernel

In the previous section, motivated by the CUDA implementation by Liu et al. (2023b), we demonstrated how activation sparsity can be leveraged for Nvidia GPUs. In this section, we shift our focus to detailed instructions on how to efficiently exploit activation sparsity on other GPUs. Given our focus on *inference* for resource-constrained devices, we will now concentrate on the efficient implementation of General Matrix Vector (GeMV) product for the Metal framework[6], a popular and widely used platform.

Figure 10a illustrates how we implement our GeMV kernel. We begin by dividing the weight matrix into tall, fixed-size tiles. Our implementation consists of three major steps:

1. Each tile is processed by a single Single-Instruction, Multiple-Data (SIMD) group, also known as a warp in CUDA programming model[7].

2. Each SIMD group (shown in a distinct color) executes a loop over the columns of its matrix tile. Each column is multiplied by a single element of the input vector, and these products are accumulated into an accumulator for each row. This method allows us to efficiently skip over columns of the matrix, conditioned on the dynamically observed sparsity of the input vector.

3. After each SIMD group completes its dot products, a final summation across different groups is performed to obtain the final result vector elements.

To verify the effectiveness of this approach, we measured the average latency of our 16-bit GeMV kernel with vector and matrix dimensions of 8192 on a MacBook Pro equipped with an Apple M2 Pro chip. As shown in Figure 10b, our implementation closely aligns with the perfect scaling (i.e., FLOPS approximation) used throughout the paper. We note that for sparsity levels higher than 90%, we observed diminishing returns, potentially due to the overhead of creating SIMD groups while skipping columns.

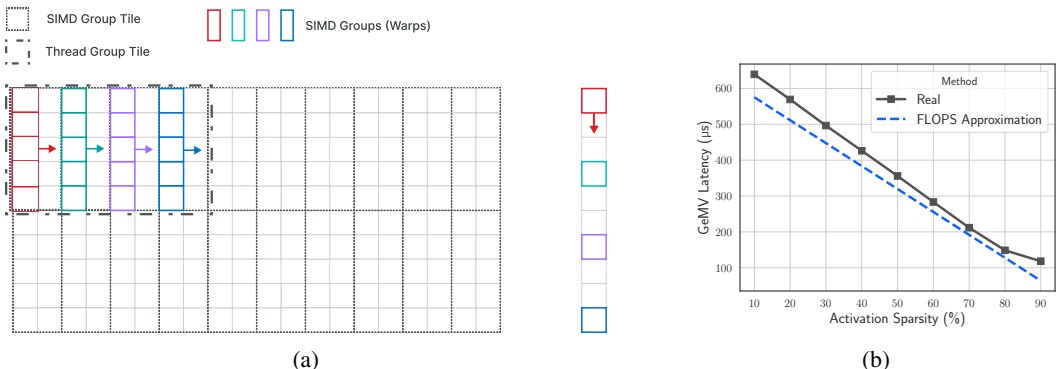

(a)                                                                                          (b)

Figure 10: **(a)** Illustration of how we implement our GeMV Kernel to exploit activation sparsity. **(b)** Performance of our GeMV implementation in comparison to the perfect scaling.

---

[6]https://developer.apple.com/metal/

[7]Technically speaking, CUDA warps follow the Single-Instruction Multiple-Threads (SIMT) paradigm, which is slightly different from the SIMD paradigm. However, for the purpose of this section, we may safely assume they are similar.

## C ACTIVATION SPARSITY AND SPECULATIVE DECODING

Speculative decoding (Leviathan et al., 2023) is a technique that uses a smaller model $M_q$ to propose $\gamma$ tokens and a larger model $M_p$ to verify those tokens and selects matching ones. This technique improves the runtime of the model by avoiding running $M_p$ sequentially per all tokens. To further improve the speculative decoding inference, we can leverage sparsity as follows.

**Latency model.** We assume a simple conceptual model for latency in speculative decoding. Following Deja Vu (Liu et al., 2023b) latency can be broken down into compute and I/O latency. The compute latency is negligible to I/O. Also, notice that the Speculative decoding is meant for the constraints that memory bandwidth is the bottleneck. Therefore we only compare I/O latency between sparse and non-sparse models. If the average aggregated sparsity of $M_p$ for $\gamma$ tokens is $\bar{s}_{\mathrm{agg}}(\gamma)$, and runtime of $M_p$ is $T$, we approximate the latency of running $M_p$ for $\gamma$ consecutive tokens with $(1 - \bar{s}_{\mathrm{agg}}(\gamma))T$. As discussed in the previous section, this is a good approximation of real latency.

**Theoretical latency improvement.** Assume the smaller model $M_q$ operates $c$ times faster than the cumbersome model $M_p$. As per the primary text, token acceptances are assumed to follow an independent and identically distributed (i.i.d.) behavior. Denote $\alpha$ as the expected probability of a token generated by $M_q$ being approved by $M_p$. The following theorems hold:

**Theorem 1.** *The expected improvement factor in latency for speculative decoding with sparsity, over standard speculative decoding is* $\frac{c\gamma+1}{c\gamma+(1-\bar{s}_{agg}(\gamma))}$.

*Proof.* The amount of time required to run sparsified model is quantified as $Tc\gamma + (1 - \bar{s}_{\mathrm{agg}}(\gamma))T$. It is the time of running a smaller model plus a larger model's non-sparse portion. Run time of speculative decoding without sparsity is $Tc\gamma + T$. The number of generated tokens in both is the same. Therefore the relative speedup of using sparsity is given by: $\frac{Tc\gamma+T}{Tc\gamma+(1-\bar{s}_{agg}(\gamma))T}$. □

**Theorem 2.** *The expected improvement factor in latency, when combining sparsity with speculative decoding against normal (autoregressive) decoding using only $M_p$, is* $\frac{1-\alpha^{\gamma+1}}{(c\gamma+(1-\bar{s}_{agg}(\gamma)))(1-\alpha)}$.

*Proof.* Similar to theorem above $Tc\gamma + (1 - \bar{s}_{\mathrm{agg}}(\gamma))T$ gives the time required for sparse speculative decoding. According to the original paper, the standard speculative decoding yields an average of $\frac{1-\alpha^{\gamma+1}}{1-\alpha}$ tokens generated per each run (Leviathan et al., 2023). Thus, the anticipated run time when generating tokens with sparsity during speculative decoding becomes $\frac{(c\gamma+(1-\bar{s}_{agg}(\gamma)))(1-\alpha)}{1-\alpha^{\gamma+1}}T$. Given the runtime for producing a single token via an autoregressive approach is $T$, the inverse of this fraction gives the desired results. □

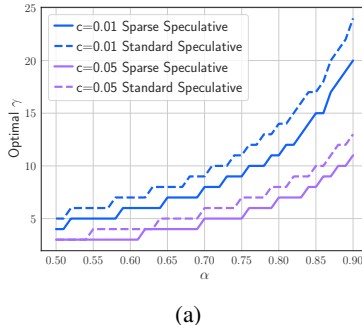
(a)

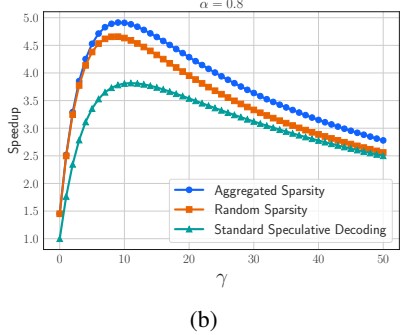
(b)

Figure 11: (a) optimal $\gamma$ for sparse speculative decoding (b) speed up of sparse speculative decoding and standard speculative decoding over autoregressive decoding when $\alpha = 0.8$ and $c = 0.02$

**Optimal $\gamma$.** The optimal $\gamma$ for speculative decoding can be found by optimizing the speedup factor equation in Theorem 2. When sparsity is not present, the equation can be solved numerically, but for reluified networks, the aggregated sparsity for different $\gamma$'s will affect the final results. We have found optimal $\gamma$s based on $\bar{s}_{\mathrm{agg}}(\gamma)$ for OPT 6.7B and presented the results in figure 11a. The chosen

$\gamma$ for sparse speculative decoding is smaller than standard speculative decoding since higher $\gamma$ will result in less sparsity. The gap in $\gamma$ is always less than 20%. Also, in figure 11b, it can be seen for the specific case of $\alpha = 0.8, c = 0.02$, the sparse speculative decoding has the highest speed-up factor over autoregressive at $\gamma = 10$s vs standard version's optimal point which happens for $\gamma = 12$. Sparse speculative decoding at $\gamma = 12$ is better than standard speculative decoding at $\gamma = 12$, while sparse speculative decoding at $\gamma = 10$ beats both. Another observation from 11b is for the case of purely random sparsity, the benefit of sparse speculative decoding would diminish over standard speculative decoding in higher $\gamma$s. In contrast, the benefits of aggregated sparsity would last for larger values of $\gamma$.

## D    PRE-ACTIVATION DISTRIBUTION OF OPT MODELS TRAINED FROM SCRATCH

The primary determinant of sparsity levels is believed to be the distribution of pre-activation inputs. As observed in Sec. 4.1, there is a significant variance in the pre-activation distributions between the Llama and Falcon models. This raises a question: if we standardize the training data and optimization algorithm, would there still be a divergence in the distribution shapes? To explore this, we trained OPT 1.3B models from the ground up using four different activation function variants. We charted the evolution of the pre-activation distributions throughout the training process in Fig. 12. Initially, these distributions are identical but they eventually begin to diverge. Moving from SiLU to ReLU (with an increasing value of $\beta$), we observe a trend towards a more centralized pre-activation distribution around zero, often approaching a uni-modal pattern. Further exploration into the dynamics of pre- and post-activation behaviors and their implications for efficiency and accuracy presents an intriguing avenue for future research.

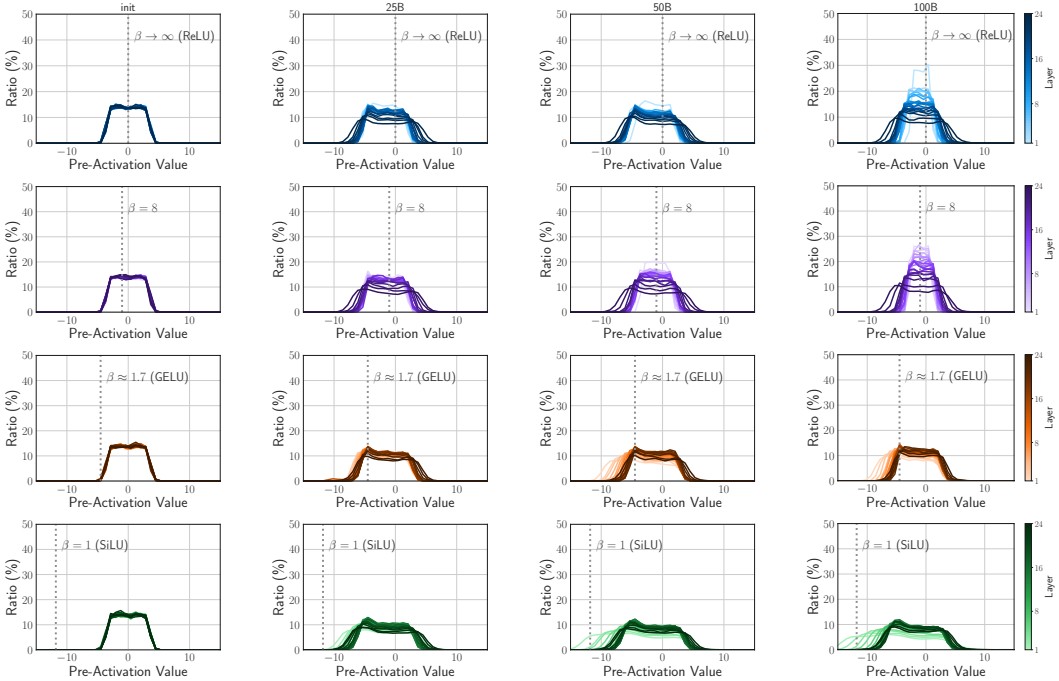

Figure 12: Pre-activation distributions of various OPT 1.3B models with all four types of activations trained from scratch at various number of seen tokens during training.

# E   IS A SPARSIFIED LARGE MODEL BETTER THAN DENSE SMALLER ONES?

When it comes to deploying efficient models, one may naturally use an original smaller-size (dense) model. The argument would be the performance of the relufied larger model might be already equal to or less than the smaller dense model. To study the above question, we plotted the performance vs. efficiency of the original and the relufied OPT models in Fig. 13. Taking the relufied OPT 6.7B model as an example, it operates at 2.8 GFLPOPs per token. Interpolating the blue line (that can be seen as a scaling plot of the OPT model), a dense model with equivalent FLOPS falls more than 2% short in zero-shot performance.

Similarly, compared to the relufied OPT 2.7B model, the equivalent (in FLOPS) dense model performs almost 2% lower. Indeed, the fact that the relufied models lie well above the scaling graph of the original OPT models, shows the effectiveness of relufication processes as a method to get better but more efficient models. As a side benefit, it makes the efficiency spectrum of the available LLMs more continuous. For example, consider a combination of hardware and use case that only allows deploying LLMs with lower than 3 GFLOPS during inference. Going with standard pretrained models, the only available option is OPT 2.7B with almost 1 GFLOPS, as the 6.7B does not satisfy the hardware constraint. In this situation, our relufied model not only falls in the limited inference budget but is also very close to the next largest available model in terms of accraucy. An exciting and timely direction for future research is finding methods, that, given an LLM (or a family of LLMs), are able to produce the best performing model matching the specified inference computation budget.

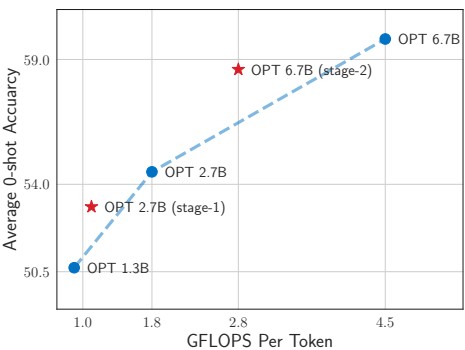

Figure 13: Performance of sparse large models vs. dense smaller models: The relufied large models (red stars) are above the scaling curve of original dense models (blue circles and dashed line).

# F   ADDITIONAL RESULTS

## F.1   EXTENDED RESULTS FOR TRAINING FROM SCRATCH

In Sec. 3, we conducted a zero-shot comparison of OPT 1.3B models that were trained from scratch. This subsection extends that analysis by presenting preliminary results from continuing the training for an additional 100 billion tokens. Our findings include:

- Prolonged training demonstrates that all models maintain comparable performance, as illustrated in Fig. 14a.
- Consistent with zero-shot accuracy results, the comparison of perplexity scores on LAMBADA reveals similar performance across all activation functions (Fig. 14b).

## F.2   COMPARING THE QUALITY OF GENERATIONS

In the main text, our focus was primarily on zero-shot and few-shot tasks, aligning our evaluation of different models with established literature standards. These tasks assess a large language model's (LLM) knowledgeability, memorization, and commonsense reasoning abilities.

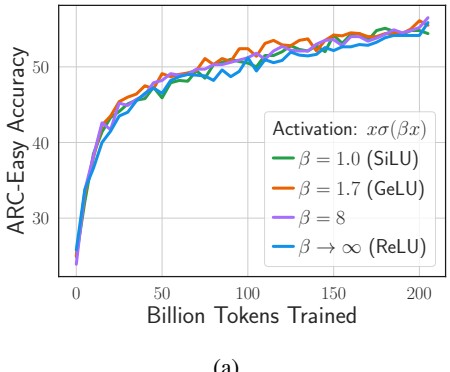
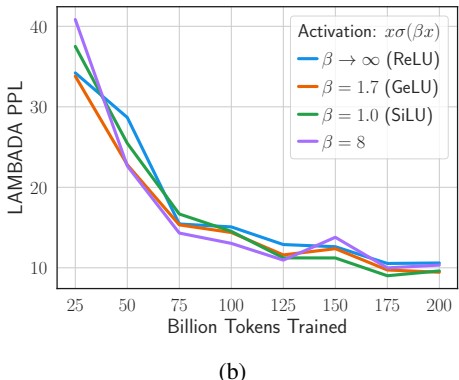

|           |           |
|:---------:|:---------:|
| (a)       | (b)       |

Figure 14: Extended training with ReLU and other activation functions shows that they continue to demonstrate competitive performance over a longer training period.

Table 3: Generation results comparison on TruthfulQA and LAMBADA: The relufication process, similar to the zero-shot scenario, does not significantly affect perplexity. Additionally, there is an observable improvement in the truthful score for most models.

| Model (stage) | FLOPS (G) | Zero-Shot Acc | TruthfulQA ROGUE1 | TruthfulQA BLEURT | LAMBADA Perplexity |
|---|---|---|---|---|---|
| Falcon 7B (s2) | 2.2 | 64.8 | 28.8 | 31.7 | 4.6 |
| Falcon 7B (s1) | 4.1 | 65.2 | 29.3 | 32.0 | 4.1 |
| Falcon 7B | 6.6 | 66.8 | 26.2 | 27.8 | 3.4 |
| Llama 7B (s2) | 2.9 | 66.4 | 26.4 | 29.1 | 4.0 |
| Llama 7B (s1) | 4.8 | 67.1 | 29.1 | 30.8 | 3.8 |
| Llama 7B | 6.6 | 68.4 | 27.1 | 29.6 | 3.5 |

This section delves into the perplexity scores of several models modified with ReLU activation, as detailed in table 3. The results indicate that, similar to the zero-shot tasks, relufication does not significantly impact the perplexity scores. Notably, in the case of TruthfulQA (Lin et al., 2022), the modified models exhibit improvements. This observation motivates further investigation into the effects of relufication on the quality of generation in other scenarios and with additional models.

In the next section, we demonstrate that the relufication process, even when assessed through perplexity scores, surpasses the performance of finetuned structured weight pruning methods.

## F.3 COMPARISON WITH PRUNING METHODS

While our work primarily focuses on *activation sparsity*, another approach to achieving sparsity is through weight pruning. In this section, we compare the performance of Llama-7B models with various weight pruning methods, including Wanda (Sun et al., 2023), SparseGPT (Frantar & Alistarh, 2023), and the magnitude pruning baseline, as shown in table 4.

Table 4: Comparison with weight pruning methods: Relufied models exhibit significantly better performance.

| Llama-7B Variant | FLOPS | Zero-Shot Accuracy | | | | | | | | | | Perplexity | |
|---|---|---|---|---|---|---|---|---|---|---|---|---|---|
| | | Avg | Arc-E | Arc-C | Hellaswag | BoolQ | LAMBADA | PIQA | TriviaQA | WinoGrande | SciQ | WikiText | LAMBADA |
| Dense (Baseline) | 100% | 68.4 | 75.5 | 42.1 | 69.9 | 74.8 | 73.1 | 78.7 | 49.9 | 69.8 | 95.4 | 9.5 | 3.5 |
| Shifted ReLU (s1) | 57% | 66.8 | 73.3 | 39.3 | 55.1 | 73.2 | 70.5 | 77.8 | 49.4 | 68.1 | 94.2 | 11.1 | 3.9 |
| ReLU (s2) | 43% | 66.4 | 73.8 | 39.6 | 54.8 | 69.9 | 70.7 | 77.9 | 48.5 | 68.6 | 93.8 | 12.8 | 4.0 |
| 2:4 weight pruning (magnitude) | 50% | 60.7 | 67.7 | 34.6 | 50.3 | 70.8 | 66.5 | 74.4 | 26.8 | 64.5 | 90.5 | 13.9 | 4.9 |
| 2:4 weight pruning (SparseGPT) | 50% | 60.1 | 66.4 | 34.2 | 50.5 | 71.6 | 68.2 | 74.6 | 19.3 | 64.4 | 91.1 | 13.6 | 4.6 |
| 2:4 weight pruning (Wanda) | 50% | 60.8 | 68.3 | 33.9 | 50.2 | 69.6 | 67.2 | 73.9 | 26.7 | 65.7 | 91.5 | 13.6 | 4.7 |

For a fair comparison, we finetuned the pruned models on the same dataset (i.e., RefinedWeb) as the relufied models. Additionally, we report results using structured pruning that are hardware-friendly, similar to activation sparsity. All models use approximately the same number of FLOPS, about half that of the dense model.

Our observations from table 4 include:

- Weight pruning methods demonstrate significantly lower zero-shot accuracy, though their perplexity scores are somewhat comparable. This aligns with the performance gap reported by Sun et al. (2023). One possible reason is that, unlike activation sparsity, which uses a small part of the model to generate a specific token, weight pruning permanently removes knowledge, reducing the computational capacity of the entire model for all tokens. This leads to a notable performance decline on certain tasks, such as TriviaQA.

- Post-finetuning, all weight pruning methods show performance similar to the magnitude pruning baseline, suggesting a challenge in recovering performance through weight pruning.

It is worth noting that further finetuning could benefit all methods. Additionally, unstructured sparsity might enhance zero-shot performance of pruning methods further. However, the practical benefits of unstructured sparsity are less clear on modern hardware, unless the level of sparsity is significantly high.

### F.4 MIXED AGGREGATED SPARSITY

In Sec. 5.1, we explored the concept of aggregated sparsity. A notable finding from that discussion is the variation in aggregated sparsity across layers: earlier layers exhibit significantly greater aggregated sparsity compared to the deeper layers, as illustrated in Fig. 7a.

This leads to an intriguing possibility: exploiting this sparsity pattern through a mixed reuse mechanism, diverging from the fixed reuse pattern discussed in Fig. 7c. For instance, with $\gamma = 4$, instead of reusing the same neurons for the next four token generations across all layers, we could reuse neurons in the earlier half of the layers for 8 token generation steps, while limiting reuse in the latter half to only 2 steps. Similarly, for $\gamma = 8$, we alternate reuse between 16 or 4 steps; for $\gamma = 16$, between 32 and 8 steps; and for $\gamma = 32$, between 64 and 8 steps[8].

As demonstrated in Fig. 15, this mixed reuse approach improves performance (i.e., reduces perplexity) when $\gamma$ is small. However, as $\gamma$ increases, the benefits diminish, and at $\gamma = 32$, the fixed policy actually outperforms the mixed approach. This may be attributed to the performance decline due to shorter reuse periods in the later layers not sufficiently compensating for the extended reuse duration in the earlier layers. However, for smaller $\gamma$ values, the reuse frequencies of the earlier and later layers are more closely aligned, leading to similar performance outcomes.

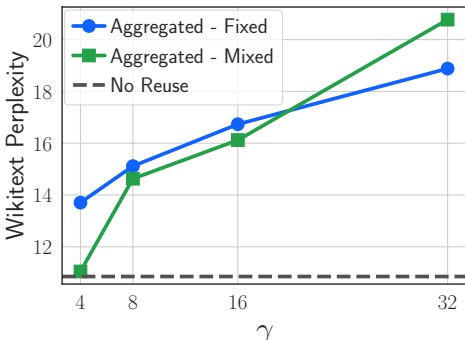

Figure 15: Comparison of mixed versus fixed aggregated sparsity: The mixed reuse policy initially outperforms the fixed approach, but for larger values of $\gamma$, the fixed policy yields better results.

---

[8]While this current mixed reuse policy was the main focus, we also empirically explored other mixed reuse combinations, which did not yield significant improvements.

