# OpenReview forum: "ReLU Strikes Back: Exploiting Activation Sparsity in Large Language Models"
_ICLR.cc/2024/Conference — ICLR 2024 oral_

### Official Review · Reviewer_iJ3a · 2023-10-31

**Soundness:** 2 fair
**Presentation:** 2 fair
**Contribution:** 2 fair
**Rating:** 6
**Confidence:** 4

**Summary:**

This paper studies the sparsity properties associated with activation functions in Large Language Models (LLMs). It highlights that LLMs employing ReLU-based activations exhibit pronounced sparsity in FFN, which can be harnessed for more efficient LLM inference. Conversely, modern LLMs commonly utilize activation functions such as GeLU and SiLU, which generate non-zero outputs even for negative inputs. This behavior hinders the model from achieving optimal sparsity. This paper proposes to replace GeLU, SiLU with ReLu for better sparsity. The paper studies both training the model from scratch and fine-tuning the model to make non-ReLU models adapt to ReLU activations. The authors suggest that replacing other activations with ReLU does not largely hurt the model performance. Finally, this paper also discusses the potential applications of this sparsity property.

**Strengths:**

1. This paper conducts an evaluation to study how different activations influence model performance under both pre-training and finetuning scenarios.
2. The evaluation of inserting ReLU in attention layers is interesting (Stage 2). Even it generally hurts in-context learning (ICL) performance.

**Weaknesses:**

1. The observation that replacing activation functions like GeLU, SiLU with ReLU only marginally influences performance is not new. It is also mentioned and discussed in [1][2].

2. The evaluation (Table 1 & 2) majorly focuses on zero-shot learning and ICL scenarios. Although I understand that zero-shot learning and ICL are common settings to compare LLM performance, it can be helpful to compare the model performance on generation tasks to better understand how different activations influence model performance.

3. I am confused by the applications studied in Section 5.1, what does “loading new weights” mean here? Shouldn’t all weights be pre-loaded to the GPU HBM in common inference frameworks like vLLM [3]?

4. Although the paper claims that replacing other activations with ReLU only has negligible impacts on the performance, the accuracy drop seems not to be so marginal. However, I agree that this replacement can also be a potential good trade-off between model performance and model efficiency.


[1] Li, Zonglin, et al. "The Lazy Neuron Phenomenon: On Emergence of Activation Sparsity in Transformers." The Eleventh International Conference on Learning Representations. 2022.

[2] Zhang, Zhengyan, et al. "MoEfication: Transformer Feed-forward Layers are Mixtures of Experts." Findings of the Association for Computational Linguistics: ACL 2022. 2022.

[3] Kwon, Woosuk, et al. "Efficient Memory Management for Large Language Model Serving with PagedAttention." Proceedings of the 29th Symposium on Operating Systems Principles. 2023.

**Questions:**

See weakness.

---

> ### Author Response · Authors · 2023-11-18
> **Response to Reviewer iJ3a [part 1]**
>
> We thank the reviewer for their constructive and valuable feedback which improves our work. Please see our detailed response below:
>
> > The observation that replacing activation functions like GeLU, SiLU with ReLU only marginally influences performance is not new. It is also mentioned and discussed in [1][2].
>
> We appreciate the reviewer's observation regarding the impact of replacing activation functions like GeLU and SiLU with ReLU. While the marginal influence of such replacements on performance has indeed been noted in previous works, our study diverges in both scope and focus:
>
> * **Scale and Scope of Study**: Our research extends beyond the replacement of activation functions in existing models, conducting a comprehensive investigation across various activation functions, models, and evaluation tasks. This significantly larger scale study is particularly noteworthy, as previous works (e.g., [1], [2]) have primarily focused on smaller-scale experiments, such as replacing GELU with ReLU in ViT or BERT. In contrast, we train 1 billion parameter models on over 200 billion tokens, thus providing a more robust and extensive evaluation on various activation functions, architectures and tasks.
> * **Inference Efficiency of LLMs**: The central motivation of our work is to analyze the implications of different activation functions on the inference efficiency of large language models (LLMs). Our findings demonstrate that commonly used activation functions do not offer a favorable balance between performance gains and efficiency or latency losses during inference. We argue in the paper that with additional one-time cost of longer training on relufied model one can get speed up during inference.
> * **Exploration of Activation Function Consequences**: Our work delves into understanding the broader consequences of using different activation functions in LLMs. For instance, we investigate the pre-activation distribution, as shown in Figure 5, which leads us to explore a rarely used activation function for LLMs: the shifted ReLU. This exploration reveals that shifted ReLU can significantly increase sparsity while maintaining competitive performance, a finding detailed in Section 5.3 and Appendix F.
> * **Practicality in Pretrained LLMs**: We emphasize the practicality of modifying activation functions in already pretrained LLMs. Our study illustrates that fine-tuning these models with ReLU does not result in a significant drop in accuracy, highlighting a trade-off that has been underexplored in prior research.
>
>
> In summary, while acknowledging the contributions of previous works, our study offers a unique perspective by examining the consequences of activation function selection in LLMs from an “inference efficiency” standpoint. We believe that our comprehensive approach and the novel insights generated, particularly in the context of model efficiency and practical adaptability, represent a significant advancement in the field.
>
> ----
>
> > The evaluation (Table 1 & 2) majorly focuses on zero-shot learning and ICL scenarios. Although I understand that zero-shot learning and ICL are common settings to compare LLM performance, it can be helpful to compare the model performance on generation tasks to better understand how different activations influence model performance.
>
> Thank you for your valuable comment. In Appendix F.2, we have included new results that compare the generation quality of several relufied models. Overall, similar to the zero-shot scenario, the relufication process does not significantly impact perplexity. More interestingly, the TruthfulQA scores of the models seem to improve in the majority of cases, which encourages us to further investigate the impact of relufication on generation quality for other scenarios and additional models. In addition, we have included additional perplexity scores throughout Appendix F (e.g., Tab.4, Fig.14b) that are in line with our conclusions on zero-shot tasks.
>
> We thank the reviewer again and hope that these enhancements to our empirical evaluations, inspired by your suggestion, will strengthen the paper.

---

> > ### Author Response · Authors · 2023-11-18
> > **Response to Reviewer iJ3a [part 2]**
> >
> > > I am confused by the applications studied in Section 5.1, what does “loading new weights” mean here? Shouldn’t all weights be pre-loaded to the GPU HBM in common inference frameworks like vLLM [3]?
> >
> > We apologize for the confusion. The reviewer is indeed correct in their assumption that weights are pre-loaded into GPU memory. However, our research highlights an additional I/O cost incurred during the token generation phase. This cost arises from transferring weights between GPU memory and computational caches and registers.
> > Our findings on aggregated sparsity, as shown in Figure 7, reveal that this cost can be significantly reduced through neuron reuse. By reusing neurons, the total weight transfer volume during multiple token generations is substantially lower compared to scenarios without reuse.
> > Furthermore, our findings have implications beyond this initial scenario. In cases where the entire model cannot fit in the GPU memory, but only a smaller subset can (i.e., by sharding or offloading), aggregated sparsity can be beneficial. For example, we can utilize a portion of the down projection layer's loaded weights for several subsequent generation steps without a significant loss in the quality of generation, as detailed in Figure 7(c).
> > We have updated our manuscript for clearer articulation of these points. We believe these findings open up intriguing optimization opportunities for exploration.
> >
> > ---
> > > Although the paper claims that replacing other activations with ReLU only has negligible impacts on the performance, the accuracy drop seems not to be so marginal. However, I agree that this replacement can also be a potential good trade-off between model performance and model efficiency.
> >
> > We agree with your assessment of the favorable trade-off between model performance and efficiency. The performance gap with ReLU activation is relatively minor and can be offset by slightly prolonged training. Considering the widespread application of current LLMs, this additional training and fine-tuning is a justified investment for the efficiency gains achieved, as illustrated in Appendix B. These gains are particularly valuable in practical scenarios, highlighting the relevance of our approach in real-world deployments.
> >
> > ---
> > In conclusion, we hope we have incorporated all reviewer's feedback into the revised manuscript. We believe that these improvements have substantially strengthened the paper and hope that it warrants reconsideration by the reviewer. Should any areas remain for further improvement, we hope to engage with the reviewer in a constructive discussion during the remainder of the discussion period to further improve our work.

---

> > > ### Comment · Reviewer_iJ3a · 2023-11-21
> > > **Thanks for the response**
> > >
> > > Thank the authors for the further clarification and additional experiments. The clarification addresses most of my concerns, and the additional evaluation improves the completeness of the evaluation. Thus, I decided to increase my rating to 6.
> > >
> > > The reason that I do not further increase the rating is that, based on the presented evaluation results, I don’t feel the performance drop caused by ReLU is small enough to be ignored (this is also mentioned by Reviewer EzRV), which limits the contribution of the paper. It will also be good to revise related arguments in the paper to make statements more accurate.

---

> > > > ### Author Response · Authors · 2023-11-23
> > > > **Thanks for the constructive feedback**
> > > >
> > > > We would like to thank the reviewer iJ3a for their constructive feedback which improved our paper a lot.

---

### Official Review · Reviewer_nzdz · 2023-10-31

**Soundness:** 3 good
**Presentation:** 3 good
**Contribution:** 2 fair
**Rating:** 8
**Confidence:** 3

**Summary:**

Recent LLMs have favored non-ReLU activations like GELU and SiLU despite their higher computation because they were thought to improve performance. This paper argues that ReLU activation can match performance of non-ReLU while significantly reducing computation due to inducing sparsity. Experiments show training LLMs from scratch with different activations yields similar performance but ReLU is much more sparse. The paper proposes "relufication" - modifying pretrained non-ReLU LLMs by replacing activations with ReLU and re-finetuning. Relufied LLMs regain original performance quickly during finetuning while being 3x more sparse, reducing computation. Additional techniques like inserting ReLU after normalization layers further improve sparsity and efficiency. Analysis shows relufied ReLU LLMs reuse neurons across tokens, enabling optimizations like faster speculative decoding. Shifted ReLU aligned to preactivations can achieve even higher sparsity with minimal impact on performance. Overall, the paper advocates reinstating ReLU in LLMs for inferencing efficiency with manageable tradeoffs.

Authors also explore aggregated sparsity, which they defined as the ratio of neurons that have not been used up to processing the first t token. They show that models using RELU display up to 50% aggregated sparsity and the usage pattern is not random so only a subset of the model can be loaded up speculatively for some cases.

**Strengths:**

- The paper tackles an important issue in deep learning - how to improve the efficiency of large language models during inference. This is a very relevant topic given the large computational requirements of state-of-the-art LLMs.
- The solutions presented by authors is very simple (applying RELU activations) making it very attractive for making
- The paper proposes practical strategies like relu-fying already existing network rather than training ones from scratch. They suggest that replacing the activation functions of pretrained LLMs with ReLU is possible, and the performance can be recovered very rapidly during finetuning. This makes this approach more practical as costly pre-training can be removed.
- Authors evaluate the performance of RELU-trained networks on a realistic benchmark, testing three models on the HEML benchmark which contains a representative sample of datasets.
- Authors show that the performance of sufficiently large models trained on sufficiently large data depends heavily on compute and data, rather than the choice of the activation function. This is supported by previous work on scaling laws (Kaplan et al., 2020; Hoffmann et al., 2022)

**Weaknesses:**

- Authors should provide more empirical comparisons to other size-reduction methods to validate thesucess of their strategy. The approach they develop is not compared to any pruning methods such as [https://openreview.net/forum?id=0GRBKLBjJE, https://arxiv.org/abs/2003.03033] which could be seen as competition.
- The sparsification mechanism relies on the underlying architecture supporting the sparse BLAS operations which is not the case for some applications. It would be good to discuss this shortcoming in more detail and perhaps include latency measurements in the main text.
- Takig advantage of the sparsity-promoting property of RELU is not trivial with regular implementations. Authors do not provide the link to their implementation of the method/experiments making applying this approach quite difficult.

**Questions:**

- What are the latency speedups of this approach on the hardware it was tested on?
- What are prerequisites to make sure a user can realize the full benefits from this approach? I understand one needs a specific implementation of the NN code to take advantage of the sparsity.

---

> ### Author Response · Authors · 2023-11-18
> **Response to Reviewer nzdz**
>
> We sincerely thank the reviewer for their insightful comments. We are encouraged by their recognition of our work's importance and its practical simplicity.  We have revised our manuscript to incorporate the reviewer’s suggestions, detailing our improvements as follows:
>
> > Authors should provide more empirical comparisons to other size-reduction methods to validate thesucess of their strategy.
>
> Thank you for suggesting this interesting experiment. In Appendix F.3, we have compared relufied models with several pruning methods. To have a fair comparison, we fixed the pruned mask and finetuned these models on the same dataset as our relufied models. Furthermore, for hardware compatibility, we utilized the 2:4 pruning method.
>
> Our results show that the weight pruning methods exhibit significantly lower zero-shot accuracy across several tasks. We partially attribute this to the inherent differences between activation sparsity and weight pruning, with the latter permanently removing parts of the model and consequently reducing its overall computational capacity and knowledge.
>
> This additional result, we believe, has significantly enhanced the comprehensiveness of our study, and we thank the reviewer for their valuable suggestion.
>
> ----
>
> > - The sparsification mechanism relies on the underlying architecture supporting the sparse BLAS operations which is not the case for some applications. It would be good to discuss this shortcoming in more detail and perhaps include latency measurements in the main text.
> > - Taking advantage of the sparsity-promoting property of RELU is not trivial with regular implementations. Authors do not provide the link to their implementation of the method/experiments making applying this approach quite difficult. What are the latency speedups of this approach on the hardware it was tested on?  What are prerequisites to make sure a user can realize the full benefits from this approach? I understand one needs a specific implementation of the NN code to take advantage of the sparsity.
>
> We acknowledge that our initial manuscript did not delve deeply into hardware support, primarily to avoid redundancy with similar discussions in the literature [1].
>
> However, recognizing the importance of this aspect, we have been working on a custom GPU kernel for a while, and we have reported latency improvements on commonly accessible hardware, such as the MacBook Pro. While we are continuing to improve our implementation, details of our efficient kernel for developers familiar with GPU programming, are provided in Appendix B.1. Additionally, Figure 10(b) demonstrates how our sparse-vector dense-matrix multiplication scales with increasing levels of sparsity. This kernel is adaptable for use in other frameworks supporting Metal Performance Shaders (MPS), such as PyTorch.
>
> -----------
>
> Finally, we hope these enhancements and additions to our manuscript address the concerns raised and merit a reconsideration of the score by the reviewer.
>
>
> References:
> *[1] Liu, Zichang, et al. "Deja vu: Contextual sparsity for efficient llms at inference time." International Conference on Machine Learning. ICML, 2023.*

---

> ### Comment · Reviewer_nzdz · 2023-11-22
>
> Dear Authors, Thank you for your thoughtful review. I appreciate the results comparing to pruning methods and I am glad to see your method being competitive.
>
> Regarding the second point, I appreciate that support for sparse operation is the limitation of the field and not necessarily your work but I think it is good to call it out explicitly. I would be very keen for you to include the explanations you provided in the comment directly in the paper, perhaps in the appendix if space is an issue.
>
> I am happy to raise my score to 8 assuming this change make its way into the camera ready version.

---

> > ### Author Response · Authors · 2023-11-23
> > **Thanks for constructive feedback**
> >
> > We would like to thank the reviewer nzdz for their constructive feedback which improved our paper a lot.

---

### Official Review · Reviewer_EzRV · 2023-11-01

**Soundness:** 3 good
**Presentation:** 3 good
**Contribution:** 4 excellent
**Rating:** 8
**Confidence:** 4

**Summary:**

This paper advocates for the use of ReLU activation function in LLM. ReLU can significantly increase the activation sparsity level, leading to promising inference efficiency. The authors argue that both training from scratch with ReLU and Relufication finetuning a trained model lead to comparable performance. In addition, the authors introduce aggregated sparsity, saying that consecutive tokens will only use a subset of all neurons as well. Aggregated sparsity can be applied on top of speculative decoding to save the I/O of loading weights.

**Strengths:**

Overall, quality and clarity are solid. This work discusses the significance of the activation function from the inference efficiency perspective, which is rather under-explored but should be discussed.  The idea of a similar sparsity pattern among consecutive tokens is also novel, to the best of my knowledge.

**Weaknesses:**

The authors argue that pretraining with other activation only gives at best marginal performance, and longer training could compensate for the gap. However, I believe this argument can be better supported. The bottom row of Figure 2 only considers three downstream datasets ( maybe perplexity would be more indicative here), and it seems like the accuracy is still growing. It is hard to judge whether longer training could compensate, and if yes, how much more training we need.

**Questions:**

(1)	We observe different aggregated sparsity ratios at different layers: the deep layer seems less sparse, according to Figure 7(a). Then, for the perplexity experiment with aggregated sparsity, did the authors use the same γ for all layers? If not, could this help recover more performance?
(2)	Could the authors elaborate on how they would imagine optimizing the finetuning process in relufication further to recover the full performance?

---

> ### Author Response · Authors · 2023-11-18
> **Response to Reviewer EzRV**
>
> We thank the reviewer for their valuable comments and feedback. We are glad to hear the reviewer found the quality and clarity of our work solid. Below, we address the comments and questions by the reviewer:
> > The authors argue that pretraining with other activation only gives at best marginal performance, and longer training could compensate for the gap. However, I believe this argument can be better supported. The bottom row of Figure 2 only considers three downstream datasets ( maybe perplexity would be more indicative here), and it seems like the accuracy is still growing. It is hard to judge whether longer training could compensate, and if yes, how much more training we need.
>
> Thanks to this comment, we have added two results in Appendix F.1 strengthening our claim. First, we have expanded our analysis to include results for up to 200 billion training tokens (an increase of 100 billion) in Fig. 14.a. Second, Fig. 14.b illustrates that the performance is not significantly impacted by the choice of activation function, even when considering perplexity on LAMBADA dataset. Our findings align with scaling laws, suggesting that longer training is indeed capable of compensating for any marginal performance gaps. We hope these added results offer a convincing argument.
> Should the reviewer have suggestions for further experiments to improve our work, we are open to incorporating them.
>
> -------
>
>
> > Could the authors elaborate on how they would imagine optimizing the finetuning process in relufication further to recover the full performance?
>
> This is an interesting question. There are several directions to further optimize the finetuning process:
>
> * Increasing the number of tokens used for finetuning could further minimize the performance gap: This extension would allow the model more time to adapt to the new activation function while retaining the learned knowledge. To make this more practical, we can explore efficient finetuning methods such as low-rank methods (e.g., LoRA).
> * More tailored training regimes: Currently, we follow the pretraining recipe of our base models. However, we believe that for short finetuning periods, we can use larger learning rates.
> * Exploring the impact of finetuning data distribution: One interesting hypothesis is that finetuning on data with a similar distribution to the training data can help stabilize the training, reduce the impact of catastrophic forgetting, and help the model recover performance faster at larger learning rates.
> * Another option may be to relufy layer by layer instead of doing it for all layer at once. This will lead to smaller changes which give the network easier time to adapt and recover.
>
> We hope this answers the reviewer's question. We are eager to know the reviewer's thoughts on these future directions
>
> -------
> > We observe different aggregated sparsity ratios at different layers: the deep layer seems less sparse, according to Figure 7(a). Then, for the perplexity experiment with aggregated sparsity, did the authors use the same γ for all layers? If not, could this help recover more performance?
>
> In our initial approach, we used a fixed gamma across all layers. However, based on your suggestion, we experimented with a mixed reuse strategy in Fig. 15 Appendix F.4. This strategy employs varying gamma values for different layers, with promising initial results. As gamma increases, we observe that shorter reuse periods in later layers do not entirely compensate the extended reuse duration in earlier layers. This insight has led us to believe that more sophisticated mixing strategies could be a fruitful area for future research. We appreciate your suggestion and believe it has significantly contributed to the depth of our study.
>
> -------
> We hope these improvements adequately address the questions raised by the reviewer and we hope that the reviewer considers the extent of these improvements in their evaluation.

---

> > ### Comment · Reviewer_EzRV · 2023-11-19
> > **Reply to the Author Response**
> >
> > Thanks to the authors for the additional experiments and discussion on the further directions.
> >
> > I think in Figure 14(b), ReLU still has a higher perplexity compared to the others. I would encourage the authors to evaluate the model with more benchmarks so that we can understand the method better.
> >
> > Nevertheless, I believe the additional experiment further supports the claim, and this work does provide a rather refreshing perspective on inference efficiency.  Thus, I would increase my score.

---

> > > ### Author Response · Authors · 2023-11-19
> > > **Response to Reviewer EzRV**
> > >
> > > We appreciate the reviewer's reconsideration and are pleased to learn
> > > that our work is regarded as offering a fresh perspective on inference
> > > efficiency. For our next revision, we will incorporate the reviewer's
> > > suggestion and report additional benchmark results.

---

### Meta-Review · Area_Chair_4uBk · 2023-12-20

**Metareview:**

The reviewers unanimously recommended accept, and appreciate that this paper focus on an important problem and proposed a solid approach.

**Justification For Why Not Higher Score:**

N/A

**Justification For Why Not Lower Score:**

This paper provides a creative solution and a solid study on a timely important problem.

---

### Decision · Program_Chairs · 2024-01-16

Accept (oral)